# Phage Display Revealed the Complex Structure of the Epitope of the Monoclonal Antibody 10H10

**DOI:** 10.3390/ijms251910311

**Published:** 2024-09-25

**Authors:** Daniil V. Shanshin, Sophia S. Borisevich, Olga N. Shaprova, Valentina S. Nesmeyanova, Alexander A. Bondar, Yuri B. Porozov, Edward M. Khamitov, Evgeniia A. Kolosova, Arseniya A. Shelemba, Nikita D. Ushkalenko, Elena V. Protopopova, Artemiy A. Sergeev, Valery B. Loktev, Dmitriy N. Shcherbakov

**Affiliations:** 1State Research Center of Virology and Biotechnology VECTOR, Rospotrebnadzor, Koltsovo 630559, Russia; dnshcherbakov@gmail.com (D.N.S.); ngelya209@gmail.com (O.N.S.); nesmeyanova_vs@vector.nsc.ru (V.S.N.); kurchanovaea@gmail.com (E.A.K.); ushkalenko@yahoo.com (N.D.U.); protopopova_ev@vector.nsc.ru (E.V.P.); sergeev_aa@vector.nsc.ru (A.A.S.); valeryloktev@gmail.com (V.B.L.); 2Laboratory of Physical and Chemical Methods of Analysis, Ufa Institute of Chemistry UFRS RAS, Ufa 450054, Russia; khamitovem@gmail.com; 3Synchrotron Radiation Facility—Siberian Circular Photon Source “SKlF” Boreskov Institute of Catalysis of Siberian Branch of the Russian Academy of Sciences, Koltsovo 630559, Russia; 4Research Institute of Biological Medicine Center for Recombinant Technologies, Altay State University, Barnaul 656049, Russia; 5Genomics Core Facility, Institute of Chemical Biology and Fundamental Medicine SB RAS, Novosibirsk 630090, Russia; alex.bondar@mail.ru; 6Laboratory of Angiopathology, The Institute of General Pathology and Pathophysiology, 8 Baltiyskaya Street, Moscow 125315, Russia; yuri.porozov@gmail.com; 7Advitam Laboratory, 11108 Belgrade, Serbia; 8Federal State Budgetary Scientific Institution “Federal Research Center for Fundamental and Translational Medicine”, Novosibirsk 630117, Russia; arseniya.shelemba@mail.ru

**Keywords:** monoclonal antibody, chimeric antibody, flavivirus, phage display, recombinant proteins

## Abstract

The annual number of reported human cases of flavivirus infections continues to increase. Measures taken by local healthcare systems and international organizations are not fully successful. In this regard, new approaches to treatment and prevention of flavivirus infections are relevant. One promising approach is to use monoclonal antibody preparations. The mouse mAb 10H10 is capable of interacting with viruses belonging to the genus *Orthoflavivirus* which are pathogenic to humans. ELISA and molecular modeling data can indicate that mAb 10H10 recognizes the fusion loop region of E protein. The KD of interaction between the mAb 10H10 and recombinant analogs of the E protein of the tick-borne encephalitis (TBEV), Zika (ZIKV) and dengue (DENV) viruses range from 1.5 to 4 nM. The aim of this study was to map the epitope of this antibody using phage display technology. After three rounds of biopanning, 60 individual phage clones were chosen. The amino acid sequences of the selected peptides were conveniently divided into five groups. Based on the selected peptides, bacteriophages were obtained carrying peptides on the surfaces of the pIII and pVIII proteins, which were tested for binding to the antibody in ELISA. Thus, the epitope of the mAb 10H10 is the highly conserved region 98-DRGWGNXXGLFGK-110 of the flavivirus E protein. The structures of the complexes of the identified peptides with the antibody paratope are proposed using the molecular docking and dynamics methods.

## 1. Introduction

The problem of flavivirus infections transmitted by arthropods (mosquitoes, ticks, etc.) has recently become increasingly urgent. There is a steady annual increase in the number of cases [1,2,3]. The family Flaviviridae is represented by four genera, including 89 known species of flaviviruses. A feature of these viruses is the presence of a similar genome structure and high similarity of protein structures, which is the cause of immunological cross-reactivity [4,5]. Flaviviruses (genus *Orthoflavivirus*) include viruses that are pathogenic to humans such as the dengue virus (DENV1–4) [6], the Zika virus (ZIKV) [7], the yellow fever virus (YFV) [8], the West Nile virus (WNV) [9], the Japanese encephalitis virus (JEV) [10] and the tick-borne encephalitis virus (TBEV) [11].

The course of the disease caused by the aforementioned flaviviruses is characterized by a wide range of clinical symptoms, ranging from high fever, headache, nausea and vomiting to myalgia and arthralgia. In some cases, the infection could develop into a serious illness, such as hemorrhagic fever, encephalitis and meningoencephalitis, leading to death [12,13,14,15,16,17,18,19,20,21,22,23]. A major concern with flaviviruses is the phenomenon of antibody-dependent enhancement (ADE) of viral infection after vaccination and repeated infections [24,25,26,27,28].

A limited number of attenuated (YFV) and inactivated (JEV and TBEV) vaccines have been developed to prevent flavivirus infections [29,30,31,32]. The application of serum therapy or monoclonal antibody preparations can be associated with complications such as ADE. Alternative approaches are required. Targeted delivery of antiviral drugs can be among such approaches. However, targeted delivery requires the presence of targeting molecules, and the antigen-recognition domains of antibodies with characterized epitope specificity can be such molecules.

We previously described the murine monoclonal antibody 10H10, which is capable of interacting with the TBEV and a number of other flaviviruses pathogenic to humans [33]. The data obtained can indicate that mAb 10H10 recognizes the highly conserved region of the E protein fusion loop [34]. However, the exact binding site of mAb 10H10 to TBEV and other flaviviruses is unknown.

Currently, the main methods for studying the epitope specificity of antibodies are X-ray diffraction analysis [35], phage display technology [36] and the use of site-directed mutagenesis [37]. All these methods make it possible to determine the area of contact of the antibody paratope in combination with the antigen epitope. In this case, one can determine the active sites of antigen–antibody interaction. However, some of these methods, primarily X-ray diffraction, require significant time investment and certain skills in their operation. Faster and simpler methods, such as Pepscan and phage display technology can be used when determining the structure of linear epitopes [36,38,39].

This study aimed to accurately map the epitope of the mAb 10H10 using phage display technology.

## 2. Results

In our previous study, we investigated the mouse mAb 10H10 [40]. In order to more accurately identify the structure of the antibody complex with the E protein, we decided to conduct a number of additional experiments. First of all, work was carried out to clarify the strength of interaction between the studied antibody and the antigen.

### 2.1. KD Estimate for Proteins

Biolayer interferometry was used to assess the antibody’s affinity to the antigen. We measured the binding affinity of mAb 10H10 to recombinant proteins (TEF1, DEF1, ZEF1) [41]. MAb 10H10 interacts with recombinant TEF1 protein, including the DI and DII of the TBEV surface protein, with an affinity (KD) of ~1.94 nM. Similarly, mAb 10H10 strongly binds ZEF1 (DI and DII of ZIKV E protein) with a KD of ~3.93 nM and DEF1 (DI and DII of DENV E protein) with a KD of ~2.91 nM (Figure 1, Table 1). From comparing the obtained values using the Dunnett test, the difference between the values was statistically not significant (*p*-value > 0.5). This means that the affinities of mAb 10H10 for various antigens are essentially equivalent.

### 2.2. Biopanning Phage Display Library

Biopanning with a 12-mer phage peptide library (Ph.D.™-12 Phage Display Peptide Library, New England BioLabs, Ipswich, MA, USA) was used to accurately map the mAb 10H10 epitope. The peptide diversity of this library is approximately 10^9^ unique sequences. Magnetic parts conjugated to protein A were used to increase the selection efficiency (orientation of variable antibody domains). After each round of affinity selection, the resulting bacteriophage suspensions were amplified in *E. coli* strain ER2738 cells.

After the third round of biopanning, 60 individual clones were randomly selected for phage DNA isolation and subsequent Sanger sequencing.

An analysis of the p3 protein gene region corresponding to the foreign peptide of the selected phage clones revealed the variants containing amino acid residues that partially coincided in sequence with the fusion loop (shown in Figure 2). Meanwhile, a number of phagotopes contained identical sequences and were represented in multiple copies of GYAGWGNSWGLF (28/60), ALRYEGLFGAPW (10/60) and SFDRGLWGLWSM (9/60) (Figure 2).

The interaction between an epitope and an antibody can be influenced by their environment. In addition to the obtained bacteriophages containing the peptide as part of the p3 protein, we decided to obtain filamentous bacteriophages containing selected peptides as part of the p8 protein. The p88 system was used for this purpose, making it possible to obtain chimeric bacteriophages in which approximately 50% of the p8 protein contained an insert and 50% was represented by the native major surface protein. We also decided to obtain bacteriophages containing a fusion loop region with 94 RDQSDRGWGNHCGLFGKG 111 (pep FL) as an insert in the p3 and p8 proteins. All the obtained bacteriophages were amplified (Table 1) and used for ELISA.

### 2.3. Immunochemical Analysis

The interaction between the obtained bacteriophages and the mAb 10H10 was characterized by ELISA. The resulting bacteriophages in equal amounts (10^11^ PFU per well) were adsorbed in immunological plates, and an antibody was then added and developed using an anti-mouse conjugate (shown in Figure 3).

The highest-intensity signal was shown to be given by bacteriophages containing peptides pep 1, pep 2, pep 3 and pep 4 as part of the p3 protein. A positive signal was also recorded for the bacteriophage containing the pep FL fusion peptide. No signal was observed in ELISA for the pep 5 peptide, which did not contain matching amino acid motifs. For bacteriophages containing peptides in the p8 protein, the signals in the ELISA were comparable to the signal of the empty bacteriophage M13. Apparently, the peptides in the p8 environment were not available for interaction with the mAb 10H10.

### 2.4. Molecular Modeling Results

The molecular docking procedure was carried out to describe the nature of interactions between the a.a. of the mAb 10H10 variable domains and the studied peptides, as well as to estimate their binding energy. The region of the active space described in [40] was considered as the binding site. All the peptides were located in this region (shown in Figure 4), with a series of intermolecular interactions (Table 2). Molecular docking typically provides a number of docking positions among which a position (Table 2) corresponding to the minimum binding energy of the peptide and antibody is selected.

Peptide binding is accompanied by the formation of a series of hydrogen bonds between amino acid residues and additional π–π stacking interactions and salt bridges (the diagrams of the interactions of amino acid residues of peptides and mAb 10H10 variable domains are presented in Appendix A).

A comparison of the key amino acid residues obtained by phage display and those identified by molecular docking revealed some discrepancies (shown in Figure 5). However, most amino acid residues were the same (D98, R99, W101, N103, G106, L107, F108, G109, K110).

In general, the molecular docking scores correlated with the results of the interaction between peptides and bacteriophages (shown in Figure 3). Thus, the highest signal was recorded in bacteriophages containing pep 2, pep 3 and pep 4. Binding of the same peptides to the antibody was characterized by minimal binding energy values (ΔG_MM–GBSA_, kcal/mol). The native peptide pep FL was characterized by a higher binding energy, but minimal values of the docking score. Meanwhile, peptide affinity analysis occurred based on a totality of docking score and ΔG_MM–GBSA_ data. The inactive pep 5 peptide was characterized by the lowest docking score. However, the energy of binding of pep 5 to the antibody was more than 7 kcal/mol lower than the energy characterizing the binding of the native peptide and pep 1 to mAb 10H10, although the ELISA signal for pep 5 was not observed. The result of molecular docking allows one to obtain a certain static, albeit thermodynamically stable, position. In other words, the resulting docking position may be unstable dynamically. Molecular dynamics simulations would allow us to estimate the retention time of the peptide at the binding site. A series of molecular dynamics simulations of the pep 1–10H10, pep 2–10H10 (leader peptides) and pep 5–10H10 (inactive peptide) complexes were carried out to explain these contradictions.

Root mean square deviations of atoms (RMSD) in the a.a. of the antibody in both simulations not exceeding 1.5 Å was an indicator of protein equilibration already at the initial simulation stages (Appendix A).

Analysis of the RMSD graphs allowed us to note that the positions of pep 1 and pep 2 into the 10H10 binding site were equalized at 100 ns of simulation, indicating the stability of the complexes (Appendix A). A different situation was observed for the pep 5–10H10 complex and antibody: starting from 120 ns of simulation, noticeable RMSD perturbations (up to 50 Å) were recorded (Appendix A). Such large fluctuations could indicate that the position of the polypeptide changed abruptly relative to the mAb 10H10 surface. Subsequent analysis of the simulation frames confirmed this.

The interaction between the a.a. of the peptide and antibody was accompanied by the formation of up to 24 different intermolecular contacts. For the pep 1–10H10 and pep 2–10H10 complexes, a similar number of contacts was maintained until the end of the simulation, and so was the peptide position in the active space of the antibody (Figure 6A,B). In the case of the pep 5–10H10 complex, the number of intermolecular contacts dropped abruptly by 120 ns of the simulation, indicating the unstable position of peptide 5 in the active space of the antibody. Intermolecular contacts between the a.a. of pep 5 and mAb 10H10 disappeared completely by 125 ns of the simulation (Figure 6C). The peptide “leaves” the active space of the site and diffuses into the solvent. In addition, the movies visualizing the interactions of peptides 1 (pep 1–10H10.mpg), 2 (pep 2–10H10.mpg) and 5 (pep 5–10H10.mpg) with mAb 10H10 can be found in the Appendix A.

Although the starting position of simulating the pep 5–10H10 complex was characterized by low binding energy as compared to that of the pep 1–10H10 complex, the peptide position in the active site of the antibody was unstable. We can assume that the contact of pep 5 with mAb 10H10 was short-lived. The discrepancy between the outcomes of molecular docking and molecular dynamics simulations can be explained by the following: molecular docking is a static calculation, whereas molecular dynamics takes into account the mobile forces of the solvent and the dynamic nature of macromolecules. The performed molecular dynamics simulation confirmed the ELISA data. Subsequently, the constructed computer model could be used for virtual mutagenesis followed by MD simulation to quickly find stabilizing mutations.

## 3. Discussion

In this study, we decided to use the BLI method to confirm the affinity interaction between the antibody and the fusion loop region. Recombinant proteins TEF1, ZEF1 and DEF1, previously obtained using the prokaryotic expression system of *E. coli*, were employed for measurements [40]. The results obtained in this work indicate the high affinity of mAb 10H10 for target proteins. The mAb 10H10 interacted with recombinant TEF1 protein with an affinity (KD) of ~1.94 nM. Meanwhile, the affinity of the mAb 10H10 for the DENV and ZIKV proteins was similar or even higher than that of TBEV ZEF1 with a KD of ~3.93 nM and DEF1 with a KD of ~2.91 nM. Not only does the antibody exhibit cross-specificity, but it also has a high affinity for the E protein of flaviviruses. An antibody is considered high-affinity if its KD value is around 10^−9^ M or lower [42]. It can be expected that the result obtained with recombinant proteins reflects the interaction of the studied antibody with the surface protein E in the virus. And it will bind with high affinity to viral particles in the body. Since 10H10 is known to not be neutralizing, the most understandable use for it may be its selective delivery of antiviral drugs. Affinity in the range of 10^−9^ M can ensure selective accumulation of the antibody conjugate with an antiviral substance at the site of virus concentration.

The most accurate methods for studying the epitope of antibodies are those based on X-ray diffraction analysis. Such techniques make it possible to most accurately determine the area of contact of the antibody paratope in a complex with the antigen site. In addition, it is possible to obtain information about the subtle features of the interaction (e.g., involvement of carbohydrates and other non-protein molecules in the interaction) [43]. Meanwhile, simpler methods can be used to solve a number of problems. It is especially reasonable when the epitope has a linear structure. Phage display technology is suitable in this case [36,38]. Phage display technology is an accessible approach that allows epitope mapping to be carried out without the use of complex, expensive equipment. Its implementation does not require special knowledge to interpret the results. It has already been used to map the epitopes of many monoclonal antibodies [44]. In a separate experiment, we evaluated the effect of adding urea (a known chaotropic agent) to an antigen solution before sorption into the wells of an immunological plate before performing ELISA with the 10H10 antibody. It turned out that the denaturation of antigens under the action of urea did not affect the result of binding to the antibody, from which we concluded that its epitope was linear (Appendix B). After three rounds of affinity selection, 60 individual phage clones were chosen. It turned out that three rounds were enough to identify an unambiguous amino acid motif. When analyzing the amino acid sequences of foreign peptides from the selected clones, a number of variants were identified that could be divided into five groups. The most represented ones were peptides containing the sequences GYAGWGNSWGLF (28 clones, first group), ALRYEGLFGAPW (10 clones, third group) and SFDRGLWGLWSM (9 clones, fourth group), having matching motifs in the amounts of 6, 4 and 3 a.a., respectively. Meanwhile, the third group included one phagotope of LPWGRWDALFGR. It had 5 a.a. coinciding with the fusion loop (shown in Figure 2). The fifth group included phagotopes not containing a.a. motifs, coinciding with the fusion loop. It is reported that an epitope must include at least eight amino acids, but energy calculations show that five to six amino acid residues provide high binding affinity [45]. Meanwhile, some authors assume that for a linear epitope, three to five amino acid residues are sufficient for strong binding, but the risk of nonspecific binding increases [46,47]. ELISA was used to experimentally confirm the binding of selected phagotopes to the mAb 10H10. Along with the obtained phage clones carrying the peptide as part of the p3 protein, we decided to use the p88 system to increase the representation of selected peptides on the surface of M13 bacteriophages. Variants of bacteriophages containing the fusion loop sequence (RDQSDRGWGNHCGLFGKG) were used in both cases as positive controls. Analysis of the ELISA results showed that the selected peptide variants specifically interacted with the monoclonal antibody 10H10. Analysis of the results of ELISA with p3-type bacteriophages of group pep 1–pep 4 showed interaction with the mAb 10H10, with an optical signal level above 1 (Figure 3). The peptide selected as a negative control showed a signal at the level of the bacteriophage without insertion. The bacteriophage containing a fragment of the fusion loop region showed an OD signal greater than 2 (Figure 3). These results indicate that the presence of a consensus motif outside the surrounding amino acid context does not necessarily ensure the highest degree of antigen–antibody affinity. Individual amino acids, their availability and, accordingly, their environment are presumably of greater importance during the antigen–antibody interaction. The pep 1 group has six amino acid residues homologous to the fusion loop but produces the smallest optical signal compared to the other selected positive peptides. Thus, it can be assumed that the most active binding sites are located in the 98-DRGWGN-103 and 106-GLFGK-110 regions. However, interaction with these regions of the antibody paratope apparently occurs independently of each other. The epitope of the mAb 10H10 can be divided into two subepitopes, with the interaction with each of them being sufficient for a stable complex to be formed (shown in Figure 5). The observed signal in ELISA is possibly a superposition of signals from the interaction between individual sections of bacteriophage peptides and the antibody.

The absence of signals in ELISA for the p88 system is apparently due to steric hindrance of the antibody’s access to the peptides. Although the presence of the selected peptide on the bacteriophage surface increases, packaging of the p8 protein limits physical access to the large antibody molecule.

A number of theoretical approaches were used in the work to describe the construction of a geometric model of the interaction of selected peptides. According to the results of molecular docking of peptides, probable interacting amino acid residues in the peptides were identified (Table 2). Comparing the results of the phage display and molecular docking, we can speak of a high degree of correspondence (Figure 5).

All the peptides used in the modeling were located in the antigen-binding region of the antibody. The set of the involved individual amino acids differed for each peptide, with occasional overlap (shown in Figure 7). Meanwhile, the potential energy of interaction indicated the possibility of the formation of a stable complex. It is interesting that low binding energy values for the complex were also obtained for pep 5. However, the result of molecular docking allowed us to obtain a certain static, albeit thermodynamically stable, position. Therefore, molecular dynamics simulations were performed to estimate the retention time of the peptide at the binding site.

The results of analyzing the contacts in the pep 1–10H10 and pep 2–10H10 complexes throughout the MD simulations allowed us to identify amino acid residues with which contact was the most prolonged (shown in Figure 8). It is interesting that comparing these results with key a.a. obtained after the phage display allowed us to detect the overlapping amino acid residues W101, F108 and R110. Despite the fact that the other positions did not coincide, this result complemented the experimental results and convinced us of the correctness of our assumption about the structure of the epitope of the monoclonal antibody 10H10. The obtained modeling results can also be indirect evidence of our assumption about the complex structure of this epitope. Peptides containing two regions, subepitope 1 and subepitope 2, and having known mobility, compete for interactions. Overall, the complex remains stable, but the position and conformation of a peptide may fluctuate, remaining in dynamic equilibrium.

## 4. Materials and Methods

### 4.1. Biolayer Interferometry

Measurement of the binding kinetics for mAb10H10 was performed on an Octet K2 instrument (ForteBio, Fremont, CA, USA) using NTA biosensors (Cat. #18-5101). Biosensors were loaded with each recombinant protein (TEF1, ZEF1, DEF1) at a concentration of 10 μg/mL for 300 s. Kinetic analysis was performed with a baseline (60 s) in PBS, an association with monoclonal antibody 10H10 (30 μg/mL), followed by two-fold dilution in PBS for 180 s and dissociation for 320 s in PBS. The monoclonal antibody 9E2 (30 mkg/mL) that did not interact with recombinant proteins (TEF1, ZEF1, DEF1) was used as a negative control. As a positive control, we also used the monoclonal antibody 9E2, which neutralizes the West Nile fever virus and interacts with the DIII of E protein. The affinity of this antibody was around 10^−9^ M. Correction for baseline drift was performed by subtracting the average of the shifts recorded by the sensor loaded with the antibody not reacting with the antigens. The data were processed using the Data Analysis HT 12.0.1.55 software. The values obtained by the control sensor were read from all other results, including the highest value of the background signals. To do this, when processing the results, a reference sensor was selected, and reference wells for subtraction were also selected. Crooked competitors were the result of dissociation. The experimental model was used in a 1:1 ratio.

### 4.2. Biopanning of the Phage Display Library

Affinity selection of the phagotopes interacting with the mouse monoclonal antibody 10H10 from the phage peptide library (Ph.D.™-12 Phage Display Peptide Library, NEB, Ipswich, MA, USA) was produced according to the manufacturer’s instruction manual.

Three rounds of affinity selection were performed. For the reaction, 5 μL of 50% aqueous solution of magnetic particles (conjugated to proteins A and G) was used. Each round included a 10-fold wash with TBST (0.15 M NaCl; 0.02 M Tris-HCl, 0.1% Tween 20, pH 7.4) in a volume of 1 mL. For the procedure, 300 ng of mouse antibody, 5 μL of magnetic particles and 10 μL of the library were mixed in TBST buffer and incubated for 20 min at room temperature with occasional stirring. Elution was performed with 1 mL of elution buffer (0.2 M glycine-HCl pH 2.2 + 1 mg/mL BSA) followed by the addition of 150 μL of neutralization buffer (1 M Tris-HCl, pH 9.1). After the first two rounds, the eluates were used for bacteriophage amplification. The titers of the bacteriophage suspensions were determined.

Amplification of the eluate was carried out by infecting *E. coli* strain ER2738 cells with a suspension of bacteriophages. Cells were cultured for 4–5 h, followed by the isolation of bacteriophages from the purified supernatant using 20% PEG/2.5 M NaCl.

As a control for the affinity selection procedure, streptavidin was used as a target according to the manufacturer’s recommendations. After three rounds of affinity selection, the HPQ motif was obtained.

### 4.3. Determination of the Infectious Titer of Bacteriophages

Titration of the bacteriophage suspensions after elution of each round of biopanning was carried out to the seventh degree of dilution; amplifications occurred after the first two rounds, to the twelfth degree of dilution. The diluted preparations were mixed with a cell culture of *E. coli* strain ER2138 (OD600 0.5); after adding 0.5% agarose, they were sown on a plate with a solid nutrient medium. The dishes were incubated at 37 °C for 16 h. The following formula was used to determine the infectious titer:C = (n × f)/V,
where C is the phage titer in plaque-forming units (PFU) per 1 mL; n is the number of plaques on plates; f is the dilution factor for this plate; and V is the volume used for titration per mL.

### 4.4. Phage DNA Sequencing

To isolate DNA, individual clones were cultured at 37 °C with constant stirring for 4.5–5 h. In order to precipitate phage particles from the suspension after separation of the *E. coli* sediment, 20% PEG/2.5 M NaCl was added and incubated overnight at 4 °C. After supernatant separation, the phage pellet was mixed with iodide buffer (10 mM Tris-HCl (pH 8.0), 1 mM EDTA, 4 M NaI) and 96% ethanol. After incubation, the DNA preparation was precipitated by centrifugation; the precipitate was washed with 70% ethanol, and centrifugation was repeated. The resulting precipitate was dissolved with water.

A typical 40 µL Sanger reaction contained 300 fmol plasmid DNA, 20 pmol DNA-specific primer (96III), 1× sequencing buffer and 2 µL BigDye v.3.1 reagent (Applied Biosystems, Forster City, CA, USA). The reaction temperature profile consisted of an initial melt at 95 °C for 2 min, followed by 50 cycles of incubation at 95 °C for 25 s, annealing at 50 °C for 5 s and elongation at 60 °C for 4 min. Sanger reactions were purified from unincorporated fluorescently labeled ddNTPs by centrifugation through individual Sephadex G-50 Fine columns (GE HealthCare, Chicago, IL, USA) (900 g, 2 min). The Sanger reaction products were then evaporated to dryness, dissolved in HiDi formamide and analyzed on an ABI PRIZM 3500XL system (Applied Biosystems, Forster City, CA, USA).

### 4.5. Production of Recombinant p88 Bacteriophages

The fth1 vector (kindly provided by Prof. J.M. Gershoni) was used to obtain recombinant phagemid [48]. The phagemid was modified by inserting PCR products that included the region encoding a foreign peptide selected by biopanning. To amplify the sequence, the forward primer comprised an SfiI hydrolysis site (NEB, Ipswich, MA, USA), a nucleotide sequence encoding the inserted peptide and a p8 protein fragment, a reverse primer complementary to the p3 protein fragment with an SpeI hydrolysis site (NEB, Ipswich, MA, USA). PCR was carried out according to the standard protocol. The resulting amplicons were treated with SfiI and SpeI restriction enzymes (NEB, Ipswich, MA, USA) and inserted into the fth1 vector treated with the appropriate enzymes. The resulting vector was transformed into *E. coli* strain Dh5αF+ according to the standard protocol. Transformed cells were cultured in 5 mL of LB medium supplemented with tetracycline (20 μg/mL). The culture was grown overnight at 37 °C with shaking at 180 rpm. The culture was then centrifuged at 6000 rpm for 20 min. The supernatant was removed, and the sediment was used to isolate plasmid DNA (Qiagen, Hilden, Germany). The presence of the insertion was confirmed by Sanger sequencing using an ABI PRISM 3130XL automatic gene analyzer (Applied Biosystems, Forster City, CA, USA).

### 4.6. ELISA

ELISA was performed on Nunc high-sorption strip 96-well polystyrene plates (Thermo Scientific, Waltham, MA, USA) according to the standard method with two-stage incubation. Immobilization of monoclonal antibodies was performed in Tris-saline buffer (0.15 M NaCl; 0.02 M Tris-HCl, pH 7.4, TSB), in 100 μL, at a concentration of 2 μg/mL at 4 °C for 18 h. The wells were blocked with 1% casein in PBS. Bacteriophage M13, which did not contain a foreign insert, was used as a negative control. The phages were dissolved in blocking buffer to a final concentration of 10^11^ PFU/well, for all bacteriophages studied, then added to the wells and incubated for 1 h at 37 °C. After the incubation, conjugates of antibodies against phage M13 immunoglobulins labeled with HRP (Invitrogen, Waltham, MA, USA) were added to the wells and incubated for 1 h at 37 °C. Plates were washed three times with PBS + 0.1% Tween 20. TMB substrate (Thermo Scientific, Waltham, MA, USA) was added to the wells, and the plates were incubated for 10 min. The enzymatic reaction was stopped with 1N HCl. The optical density (OD) was measured on a VarioScan 6Lux spectrophotometer (Thermo Fisher, Waltham, MA, USA) at a wavelength of 450 nm.

### 4.7. Statistical Analysis

Statistical analyses were performed using GraphPad Prism version 9.3.1. (GraphPad Software, San Diego, CA, USA). Statistical intergroup differences were estimated using Dunnett’s multiple comparisons test (*p*-value < 0.0001—statistically significant difference compared to M13; *p*-value > 0.05 (ns)—statistically insignificant difference compared to M13). The results of the interaction between the variable domains of mAb 10H10 and the bacteriophages (M13 + pep 1, M13 + pep 2, M13 + pep 3, M13 + pep 4, M13 + pep 5 and M13 + pep FL) are presented as the average values with SD and ranges. Differences with significance at *p* < 0.05 were considered statistically significant.

### 4.8. Molecular Modeling Study

#### 4.8.1. Protein and Peptide Preparation

Molecular modeling was carried out using the geometric parameters of the mAb 10H10 variable domains whose preparation procedure was described earlier [40]. The peptides were modeled based on the TBEV virus glycoprotein loop excised from the protein structure corresponding to PDB [49] code 1SVB [50]. Model peptide structures were prepared for calculation using Schrödinger Protein PrepWizard tools(Schrödinger, Inc., New York, NY, USA)**.** Hydrogen atoms were added and minimized, missing amino acid side chains were added, bond multiplicities were restored, and solvent molecules were removed. One chain was selected in the glycoprotein. The peptide structures were optimized in the OPLS4 force field [51] at physiological pH values. Additionally, the studied peptides were placed in a virtual cubic box filled with 0.15 M NaCl saline solution, with a 20 Å buffer zone, and were minimized by performing short molecular dynamics using the Desmond routine for 2 ns, NPT ensemble, T = 310 K.

#### 4.8.2. Molecular Docking Procedure

The peptide docking procedure was carried out using the Glide plugin implemented in the Schrödinger Suite Release-2021-2 software. The site of mAb 10H10 binding to the glycoprotein loop of the ZIKV and TBEV viruses was chosen as the search space. Semi-flexible docking was performed, where the antibody structure was taken into account as a rigid body, and conformational mobility was allowed for polypeptides. The ranking of docking determinations was carried out by assessing the following calculated parameters: docking score and binding energy ΔG_MM–GBSA_ in the presence of an implicit solvent (Water) (Schrödinger, Inc., New York, NY, USA)**.** [52].

#### 4.8.3. Molecular Dynamics

Favorable peptide–antibody positions were selected to build models for molecular dynamics simulations. The resulting peptide–protein complex was placed in a cubic system, with a buffer zone 20 Å from the protein surface. The system was filled with a 0.15 M aqueous NaCl solution. Solvent model—TIP3P. Environment—NVT. The period of the recorded dynamics simulation was 150 nsec at a temperature of 310 K (37 °C). The protocol for preparing the system for simulations included the preliminary minimization and balancing of system components. Next, a clustering procedure was used to determine the geometric parameters of static significant peptide–protein complexes.

## 5. Conclusions

The mouse mAb 10H10 has a high affinity for TEF1 (KD) ~1.94 nM, ZEF1 (KD) ~3.93 nM and DEF1 (KD) ~2.91 nM. These results indicated the prospects of using its antigen recognition domains for drug development. Affinity selection has made it possible to identify a number of phagotopes having an affinity for an antibody. Peptide analysis allowed us to localize the amino acid sequence of the 98-DRGWGNXXGLFGK-110 epitope. However, the consensus motif obtained by aligning the selected peptides was somewhat different from those obtained by theoretical calculations and by using the molecular docking and molecular dynamics methods. Based on this, we hypothesized that this epitope includes two binding regions, 98-DRGWGN-103 and 106-GLFGK-110, to the antibody paratope. Perhaps, the competitive nature of the interaction of peptides possessing both of these binding centers reduces the resulting affinity for the antibody. It can be assumed that the sequence DRGWGNXXGLFGK forms a pseudolinear epitope, while its partial conformational restrictions are required for a stable complex with the antibody to be formed, thus allowing for the largest number of intermolecular contacts to be implemented. Such a detailed study of epitope structure is of fundamental importance because it allows us to provide a molecular basis for the cross-reactive nature of this antibody. A precise understanding of the amino acid residues involved in the formation of the complex allows us to state that this antibody can interact not only with the proteins of the flaviviruses used in this work, but also with a large number of others that have similar amino acid sequences of their fusion loop. In an applied sense, this work is important as a basis for changing the structure of antibodies to obtain a chimeric or humanized variant that will have reduced allergenicity. The modified antibody can become a platform for creating a pan-flavivirus antiviral drug.

## Figures and Tables

**Figure 1 ijms-25-10311-f001:**
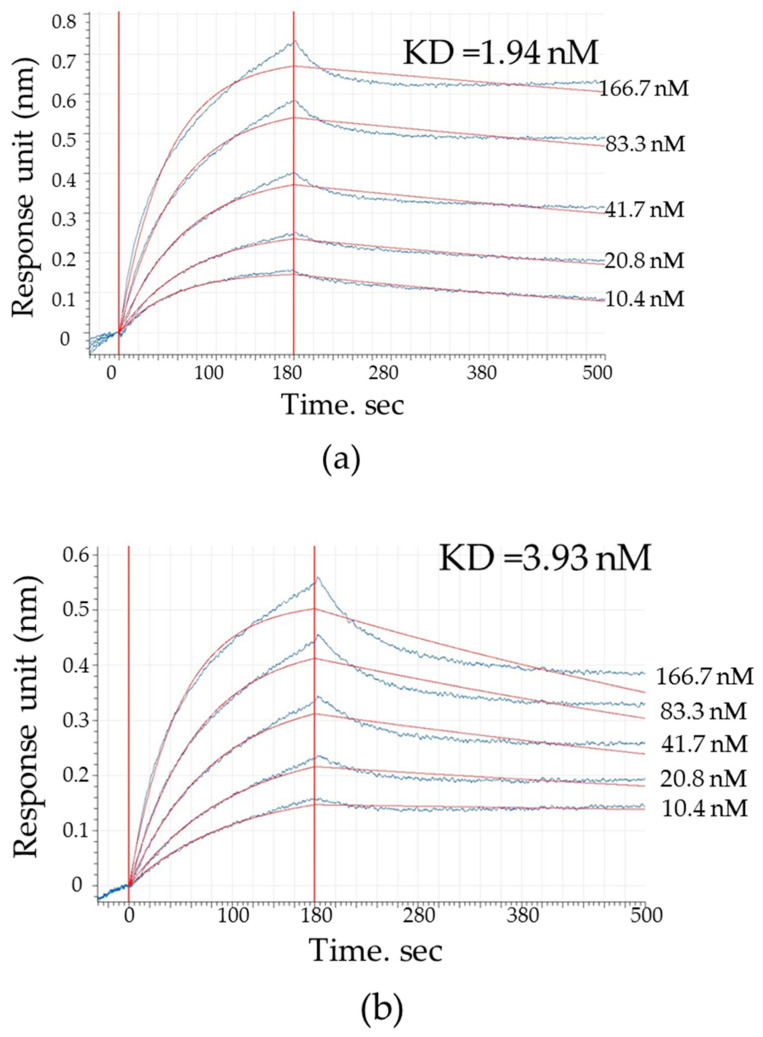
Sensorgram curves for TEF1, DEF1 and ZEF1 binding of increasing concentrations of mAb 10H10 in a BLI-kinetic assay. Biosensors were each loaded with recombinant protein (TEF1, ZEF1, DEF1) with a concentration of 10 μg/mL for 300 s. Kinetic analysis was performed with a baseline (60 s) in PBS, an association with a monoclonal antibody 10H10 for 180 s and dissociation for 320 s in PBS. The blue line indicates the values obtained from the interaction of recombinant antigens with mAb 10H10 at different concentrations (166.7 nM, 83.3 nM, 41.7 nM, 20.8 nM, 10.4 nM); the brown line indicates the average value. (**a**) Sensorgram curves for TEF1 binding of mAb 10H10; (**b**) sensorgram curves for DEF1 binding of mAb 10H10; (**c**) sensorgram curves for ZEF1 binding of mAb 10H10.

**Figure 2 ijms-25-10311-f002:**
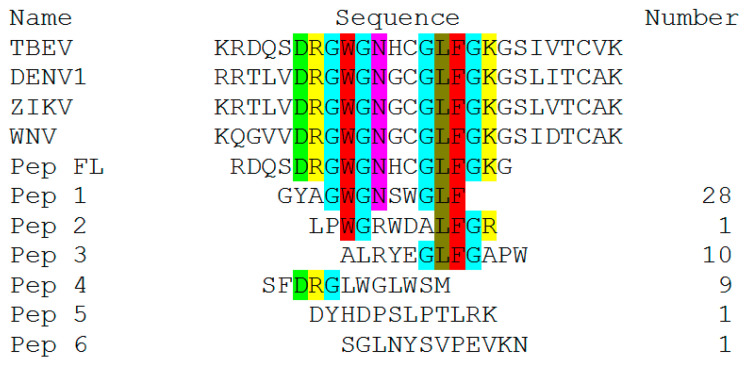
Alignment of the fusion loop of the flaviviruses and peptides with the highest enrichment scores after three rounds of affinity selection with the mAb 10H10. The negatively charged residues (D) are shown in light green; the positively charged residues (R and K) are shown in yellow and green, respectively; conformationally special residues (G) are shown in cyan; aromatic residues (W and F) are shown in red; aliphatic/hydrophobic residues (L and N) are shown in brownish green and purple, respectively. TBEV—Hypr strain (U39292.1), DENV1—Nauru/West strain Pac/1974 (U88536.1), ZIKV—MR-766 strain (KX377335.1), WNV—956 strain (NC_001563.2).

**Figure 3 ijms-25-10311-f003:**
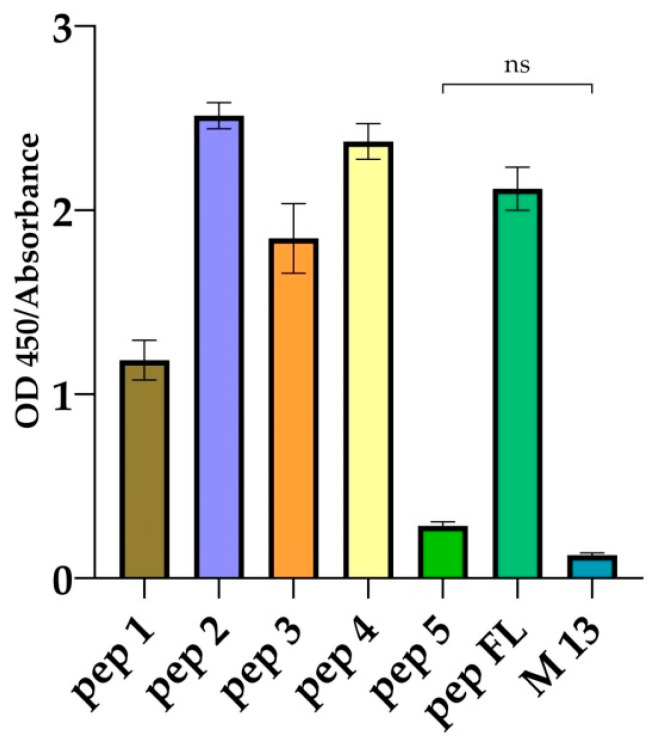
Results of the ELISA for the interaction between the mAb 10H10 and bacteriophages. Data are represented by the means ± standard error (S.E: *n =* 4) of the optical density. *p*-value > 0.05; (ns)—statistically not significant difference compared to phage M13 wild type using Dunnett’s multiple t-test. Pep 1—bacteriophage M13 containing the sequence GYAGWGNSWGLF as part of the p3 protein; pep 2—bacteriophage M13 containing the sequence LPWGRWDALFGR as part of the p3 protein; pep 3—bacteriophage M13 containing the sequence ALRYEGLFGAPW as part of the p3 protein; pep 4—bacteriophage M13 containing the sequence SFDRGLWGLWSM as part of the p3 protein; pep 5—bacteriophage M13 containing the sequence DYHDPSLPTLRK as part of the p3 protein; pep FL—bacteriophage M13 containing the sequence RDQSDRGWGNHCGLFGKG as part of the p3 protein; and M13—bacteriophage M13 not containing a foreign insert.

**Figure 4 ijms-25-10311-f004:**
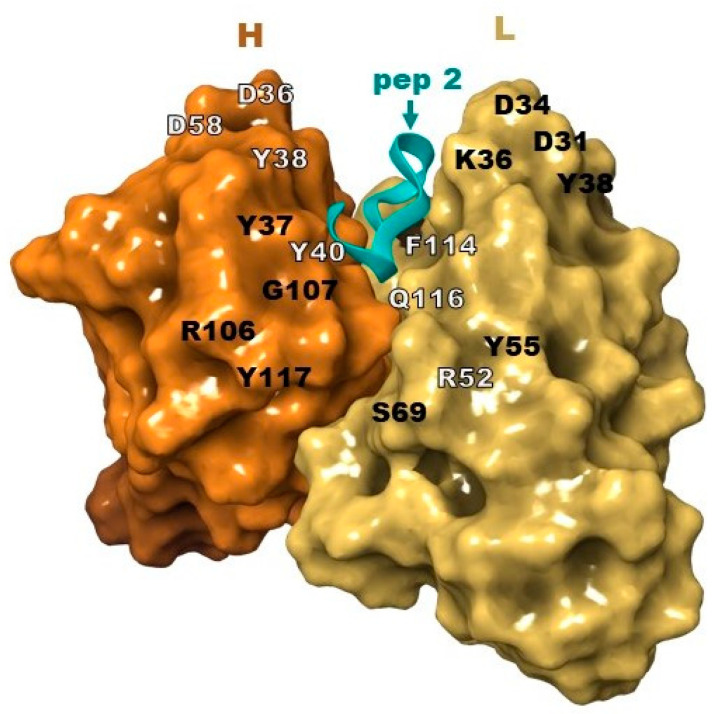
Location of pep 2 in the binding site of the mAb 10H10.

**Figure 5 ijms-25-10311-f005:**
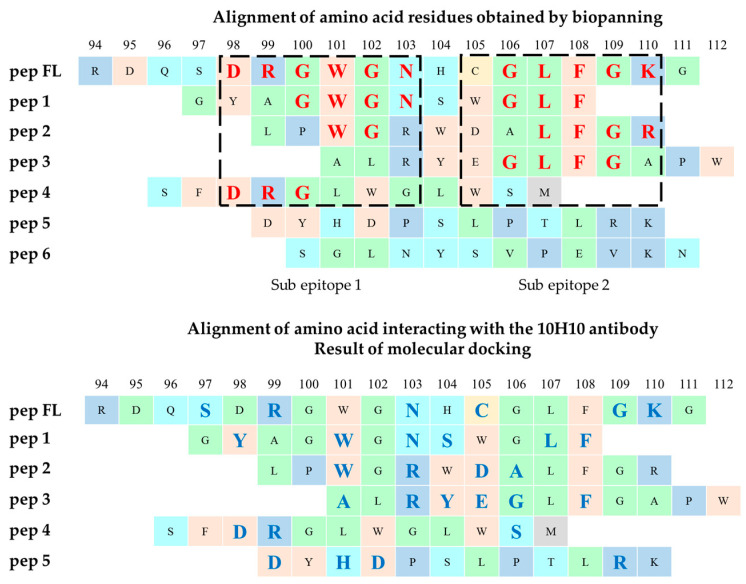
Alignment of amino acid sequences of peptides: amino acids of peptides in contact with amino acids in antibodies are highlighted in red (results from biological testing) and blue (results from molecular modeling).

**Figure 6 ijms-25-10311-f006:**
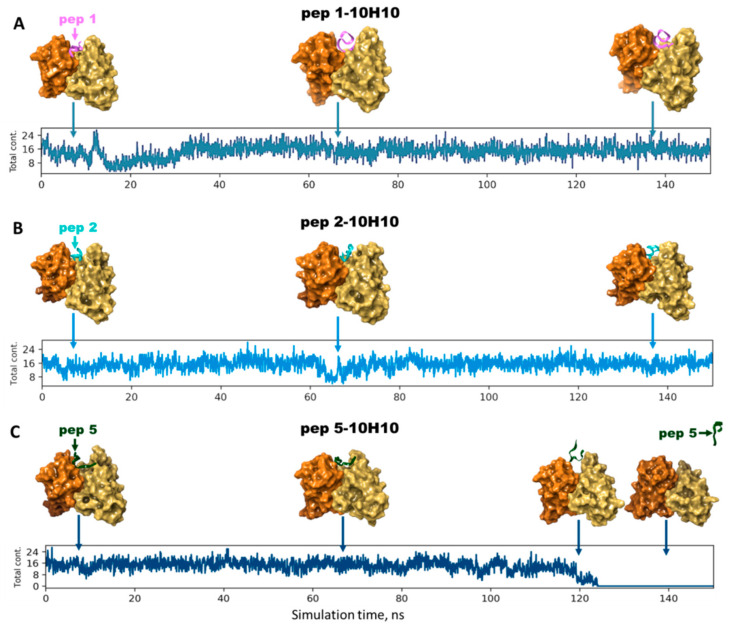
Locations of pep 1, 2 and 5 in the binding site of the mAb 10H10. (**A**)—the result of the simulation of the interaction of pep 1 with mAb 10H10, (**B**)—the result of the simulation of the interaction of pep 2 with mAb 10H10, (**C**)—the result of the simulation of the interaction of pep 5 with mAb 10H10.

**Figure 7 ijms-25-10311-f007:**
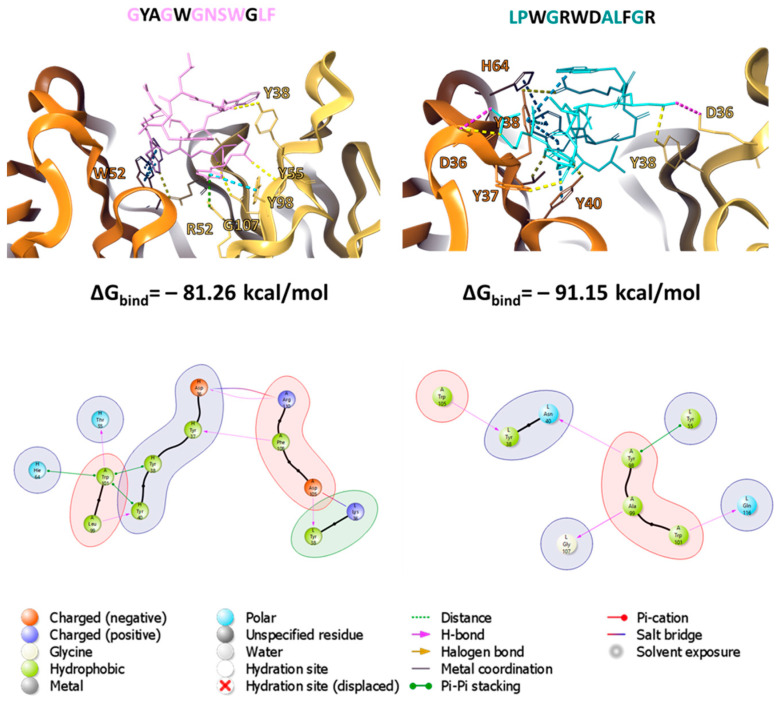
Model of the interaction of pep 1 and pep 2 with the paratope of the mAb 10H10.

**Figure 8 ijms-25-10311-f008:**
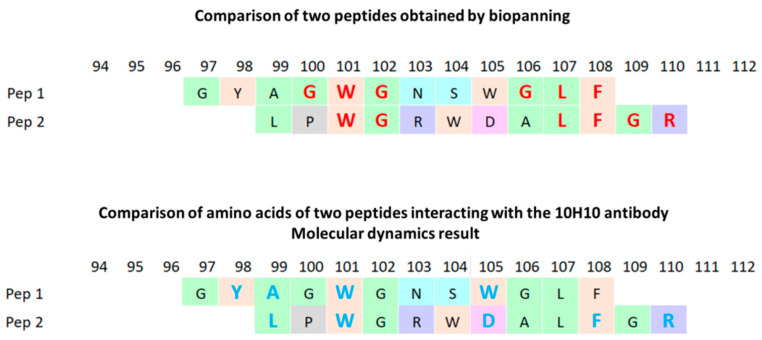
Alignment of the amino acid sequences of peptides: amino acids of peptides that contact with amino acids in antibodies are highlighted red (results of biological tests) and blue (results from molecular modeling).

**Table 1 ijms-25-10311-t001:** Types of recombinant phages.

Phagotope Name	PFU */mL	Type of Display
Pep 1	5 × 10^12^	p3
Pep 2	3 × 10^12^	p3
Pep 3	2 × 10^12^	p3
Pep 4	8 × 10^12^	p3
Pep 5	4 × 10^12^	p3
Pep FL	3 × 10^12^	p3
Pep 1	4 × 10^12^	p88
Pep 2	5 × 10^12^	p88
Pep 3	3 × 10^12^	p88
Pep 4	6 × 10^12^	p88
Pep 5	4 × 10^12^	p88
Pep FL	4 × 10^12^	p88

* plaque-forming units.

**Table 2 ijms-25-10311-t002:** Molecular docking results.

Peptide ID	Poses (Max. 100)	Docking Score	ΔG_MM–GBSA_, kcal/mol	Interaction with Amino Acids	
				H-Bonding	Other Interactions
pep Fl	34	–8.11	–65.0	H: Y37; D58 L: S32; Y38; R52; S69; W108	H: D58—salt bridge
pep 1	100	–7.52	–63.8	H: D36; Y38; W55; R106 L: K36	H: Y38—π–π stacking
pep 2	76	–6.33	–85.0	H: D36; W55; D58 L: K36; R52	H: Y38—π–π stacking; D58—salt bridge L: K36—salt bridge; F114—π–π stacking
pep 3	100	–8.39	–78.5	H: 106 L: N40; R52; G107	None
pep 4	100	–6.21	–73.9	H: D58; R106; Y117 L: G107; T108	L: R52—salt bridge
pep 5	92	–5.58	–72.0	H: D36; R106 L: K36; R52; K66; T108; H109	L: Y38—π–π stacking; K66—salt bridge

## Data Availability

Data is contained within the article and Appendix A.

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
