# Peer review of "Phage Display Revealed the Complex Structure of the Epitope of the Monoclonal Antibody 10H10"

_ijms, 2024, doi:10.3390/ijms251910311_

Round 1

Reviewer 1 Report

Comments and Suggestions for Authors

Dear authors 

The manuscript is very interesting, however I have some comments for improving the manuscript.

RESULTS SECTION 

Line 93-94.Please homogenize the term “kd” or “KD”.

line 96-97. please check  the following terms “insignificant” is it correct or talking about of not significant for a p value >̣0.5.

Line 114.Please to explain why you select three and not more selection rounds?

Figure 1. Please increase the size because the labels do not see and include subsections such as  A to C and include it in the explanation text.

In the explanation about strong binding of antibody and antigen mentioned that these interactions are strong; I consider that should include the value control with which it is compared.   

In the figure caption figure 1 has not the letter size correct of format journal.

Line 114,240. Please change the E.coli by E. coli in italics.

Figure caption 3. Please re-write the explanation about statistical significance and include the name test used.

Line 129. Please make reference to (Figure 2).

Lines 158-164. Please add the control peptide description used in immunoassays.

Lines 166-172. Please add the control positive condition  docking interaction was used in the Molecular modeling results section, it could be helped to explain the results mentioned.

Lines 186-191. 

Line 199-201. Please explain why analyze the pep1 and 2-10H10 to 100 ns and these are compared with pep-5 10H10 to 120 ns?, the perturbations to 120 ns are not observed in the pep1 and 2-10H10 to 120ns or not were evaluated.

Lines 225-269. Please rewrite this part, because repeat the results and include in discussion about phage display results compared with other studies, sensibility, specificity of technique.

Discussion section 

Table 1. Please add the meaning of the column two PFU/ml in the results description. This value is important by interpretation results interaction peptide-ab, explain it, please.

Line 261.Please explain how the urea did not affect the interaction ab-protein or antibody structure 3D affected the paratope?, because in general the interactions ab-protein are dependent on 3D conserved structure. I don't understand it, the method ELISA was used with or without UREA, please check it.

Lines 276-297. I think this part showed more results used a new propose the p88 system  and after described a modification the peptides that include a fusion loop

The discussion section should be improved, should be added studies to contrast your results with other methods, for example is it similar to results obtained from X ray diffraction in other studies?

Please discuss the results obtained about pep 5 Why for pep-5 is similar the energy binding compared with pep-1 or pep-2 showed that could be stable the interaction pep-5, however it is not waiting, please explain it.

  Method section

In the method section about ELISA Assay do not describe the use of the UREA and what concentration was used? and should include the references of previous studies.

in the Biolayer interferometry and Biopanning of phage display library described the controls used of both assays.

Please include the control description used in the method and techniques assays used. It is important to clearly explain the results.

Comments on the Quality of English Language

Minor editing of English language required.

Author Response

Thank you very much for taking the time to review this manuscript. Please find the detailed responses below and the corresponding revisions in the re-submitted files.

Comments 1: Line 93-94.Please homogenize the term “kd” or “KD”.

Response 1: Thank you for pointing this out comment. We agree with this comment. We fixed this. (line 33 p1, line 90 p.3, line 96 p.3, line 252 p.11 line 253 p.11, line 254 p.11 line 256 p.11)

Comments 2:line 96-97. please check  the following terms “insignificant” is it correct or talking about of not significant for a p value >̣0.5.

Response 2: Thank you for pointing this out comment. We agree with this comment. We replaced the word "insignificant" with "not significant " (line 98 p.3)

Comments 3:Line 114.Please to explain why you select three and not more selection rounds?

Response 3: Thank you for pointing this out comment. According to the manual the sequence of work is as follows

Since the commercial phage peptide library of the NEB was used for the affinity selection experiments, all operations were carried out according to the NEB operation manual. It recommends conducting three rounds of affinity selection. According to the recommendation, if a motif is detected in the selected phagetopes after the third round, the following rounds are not necessary.

Comments 4:Figure 1. Please increase the size because the labels do not see and include subsections such as  A to C and include it in the explanation text.

Response 4: Thank you for pointing this out comment. We agree with this comment. We fixed this (line 101-103 p.3-4)

Comments 5:In the explanation about strong binding of antibody and antigen mentioned that these interactions are strong; I consider that should include the value control with which it is compared

Response 5: Thank you for pointing this out comment. We agree with this comment. We have added text fragments to the methods and discussion sections.

An antibody is considered high-affinity if the Kd value is around 10-9 M or lower.

In our work, as a control in measurements, we also used the monoclonal antibody 9E2, which neutralizes the West Nile fever virus, the affinity of which was also around 10-9 M. (line 364-368 p14)

Comments 6:In the figure caption figure 1 has not the letter size correct of format journal.

Response 6: Thank you for pointing this out comment. We agree with this comment. We fixed this. (line 101-103 p.3-4)

Comments 7:Line 114,240. Please change the E.coli by E. coli in italics.

Response 7: Thank you for pointing this out comment. We agree with this comment. We fixed this. (line 119 p4, line 249 p11, line 390 p15, line 400 p15, line 410 p15 line 432 p16)

Comments 8:Figure caption 3. Please re-write the explanation about statistical significance and include the name test used.

Response 8: Thank you for pointing this out comment. We agree with this comment. We fixed this.Data are means ± standard error (S.E: n  4) of optical densety. p-value > 0.05 (ns) – statistically not significant difference compared to phage M13 wild type using Dunnett’s multiple t-test (line 157-159 p6)

Comments 9: Line 129. Please make reference to (Figure 2).

Response 9: Thank you for pointing this out comment. We agree with this comment. We added this, we also added references to figure 2

TBEV – strain Hypr (U39292.1), DENV1 – strain Nauru/West Pac/1974 (U88536.1), ZIKV – strain MR-766 (KX377335.1), WNV – strain 956 (NC_001563.2).(Line130-131   p5 ). …. and SFDRGLWGLWSM (9/60) (Fig.2). (line 137p.5)

Comments 10:Lines 158-164. Please add the control peptide description used in immunoassays.

Response 10: Thank you for pointing this out comment. We agree with this comment.  We added this,

pep FL - bacteriophage M13 containing the sequence RDQSDRGWGNHCGLFGKG as part of the p3 protein (line 165 p.6)

Comments 11:Lines 166-172. Please add the control positive condition  docking interaction was used in the Molecular modeling results section, it could be helped to explain the results mentioned.

Response 11: Thank you for pointing this out comment. However, it is not clear what the reviewer means by "positive control of molecular docking"? The result of molecular docking between an antibody and a FL loop could be considered a reference and possible positive control. This is mentioned in line 189 and table 2. The docking results should be interpreted as peptides binding to an antibody, but the stability of these complexes and duration of intermolecular interactions can be estimated by analyzing molecular dynamics simulations, which have been done.

Comments 12:Lines 186-191. 

Response 12: Thank you for your attention to these lines. We have found and corrected inaccuracies, and added one sentence to explain the need for molecular dynamics simulations.

Comments 13:Line 199-201. Please explain why analyze the pep1 and 2-10H10 to 100 ns and these are compared with pep-5 10H10 to 120 ns?, the perturbations to 120 ns are not observed in the pep1 and 2-10H10 to 120ns or not were evaluated.

Response 13: Thank you for pointing this out comment. We are talking about analyzing the trajectory of molecular dynamics simulations of complexes. Based on the RMSD graphs (which are given in the supplementary materials), pip1 and pip2 are located stably at the binding site throughout the entire trajectory, while the position of pep5 is unstable. The position of the peptide changes, and starting from 120, it diffuses into the solvent during the simulation. Additionally, we have rewritten this paragraph. (Line 214-218 p9)

Comments 14:Lines 225-269. Please rewrite this part, because repeat the results and include in discussion about phage display results compared with other studies, sensibility, specificity of technique.

Response 13: Thank you for pointing this out comment. We have tried to take into account the comments and rewrite this section. Line 238-241p9-10

Discussion section 

Comments 15:Table 1. Please add the meaning of the column two PFU/ml in the results description. This value is important by interpretation results interaction peptide-ab, explain it, please.

Response 15: Thank you for pointing this out comment. We have added it to the text of the article.

PFU*/ml (line 147 p5) * - plaque forming units (line 148 p6) The resulting bacteriophages in equal amounts (1011 PFU per well) were adsorbed in immunological plates, an antibody was then added and developed using an anti-mouse conjugate (shown in Fig.3). (line 150-153 p 6)

Comments 16:Line 261.Please explain how the urea did not affect the interaction ab-protein or antibody structure 3D affected the paratope?, because in general the interactions ab-protein are dependent on 3D conserved structure. I don't understand it, the method ELISA was used with or without UREA, please check it.

Response 16: Thank you for pointing this out comment. We have added an additional section (Appendix A) to the article.

In a separate experiment, we evaluated the effect of adding urea (a known chaotropic agent) to the antigen solution before sorption into the wells of an immunological plate before performing ELISA with the 10H10 antibody. It turned out that denaturation of antigens under the action of urea does not affect the result of binding to the antibody, from which we conclude that its epitope is linear (Appendix A). (line 273-278 p11)

Comments 17:Lines 276-297. I think this part showed more results used a new propose the p88 system  and after described a modification the peptides that include a fusion loop

Response 17: Thank you for pointing this out comment. We have made adjustments to this section.

Variants of bacteriophages containing the fusion loop sequence (RDQSDRGWGNHCGLFGKG) were used in both cases as positive controls. Analysis of the ELISA results showed that the selected peptide variants specifically interact with the monoclonal antibody 10H10. Analysis of the results of ELISA with p3-type bacteriophages of group 1-4 peptides showed interaction with the mAb 10H10, with an optical signal level above 1 (Fig.3). The peptide selected as a negative control showed a signal at the level of the bacteriophage without insertion. The bacteriophage containing a fragment of the fusion loop region showed an OD signal greater than 2 (Fig.3). (Line 295-300 p 11-12

Comments 18:The discussion section should be improved, should be added studies to contrast your results with other methods, for example is it similar to results obtained from X ray diffraction in other studies?

Response 18: Thank you for pointing this out comment. We have made changes to this section. Unfortunately, our results are not entirely correct to compare with the results of X-ray structural analysis.

Comments 19:Please discuss the results obtained about pep 5 Why for pep-5 is similar the energy binding compared with pep-1 or pep-2 showed that could be stable the interaction pep-5, however it is not waiting, please explain it.

Response 19: Thank you for pointing this out comment. The binding energy of peptides and antibodies was estimated for the docking positions. As a result, molecular docking would yield some static and energetically stable positions pep-5-10H10 complex relative to other peptides.

However, in addition to the thermodynamic factor, there is also a kinetic factor to consider. The low binding energy of a docking position does not necessarily mean that the peptide will remain stable in its binding site during molecular dynamics simulations. This is because molecular docking is a static calculation, whereas molecular dynamics takes into account the mobile forces of the solvent and the dynamic nature of molecules. We provide this in lines 238-241 p9-10.

  Method section

Comments 20:In the method section about ELISA Assay do not describe the use of the UREA and what concentration was used? and should include the references of previous studies.

Response 20: Thank you for pointing this out comment. We have added an additional section (Appendix A) to the article.

In a separate experiment, we evaluated the effect of adding urea (a known chaotropic agent) to the antigen solution before sorption into the wells of an immunological plate before performing ELISA with the 10H10 antibody. It turned out that denaturation of antigens under the action of urea does not affect the result of binding to the antibody, from which we conclude that its epitope is linear (Appendix A). line 273-278 p11

Comments 21: in the Biolayer interferometry and Biopanning of phage display library described the controls used of both assays.

Response 21: Thank you very much for your comments. We have added a description of the controls to the methods section.

The monoclonal antibody 9E2 (30 mkg/ml) that does not interact with recombinant proteins (TEF1, ZEF1, DEF1) was used as a negative control. As a positive control, we also used the monoclonal antibody 9E2, which neutralizes the West Nile fever virus, and interact with DIII of E protein. The affinity of this antibody was around 10-9 M. Line 364-368 p14

As a control for the affinity selection procedure, streptavidin was used as a target according to the manufacturer's recommendations. After three rounds of affinity selection, the HPQ motif was obtained. Line 393-395 p15

Comments 22: Please include the control description used in the method and techniques assays used. It is important to clearly explain the results.

Response 22: Thank you very much for your comments. This is really important information to include in the article. We have tried to include the description of the relevant controls in the text of the article.

The monoclonal antibody 9E2 (30 mkg/ml) that does not interact with recombinant proteins (TEF1, ZEF1, DEF1) was used as a negative control. As a positive control, we also used the monoclonal antibody 9E2, which neutralizes the West Nile fever virus, and interact with DIII of E protein. The affinity of this antibody was around 10-9 M. Line 364-368 p14

As a control for the affinity selection procedure, streptavidin was used as a target according to the manufacturer's recommendations. After three rounds of affinity selection, the HPQ motif was obtained. Line 393-395 p15

Reviewer 2 Report

Comments and Suggestions for Authors

Line 61 - suggest: after vaccination or repeated infections ...

Line 67 - "Characterized" may be a better word than "known".

Line 97 - suggest: affinities are essentially equivalent

General comment:  there is a great deal of information and data characterizing the activity of these antibody sets with constructed and expressed antigens. It would be beneficial to relate how this compares to and is relevant to antibody binding to native viral protein. Are there biological consequences (neutralization, etc.)?

Line 477 - suggest: delete "our experiments have shown" It is wordy and unnecessary.

This approach has defined the binding structures very well. Are there biological consequences or implication from this that relate to infectivity or tissue tropism? Why was this study important?

Comments on the Quality of English Language

The paper is well-prepared. SOme minor attention to word selection will be useful.

Author Response

Thank you very much for taking the time to review this manuscript. Please find the detailed responses below and the corresponding revisions in the re-submitted files.

Comments 1: Line 61 - suggest: after vaccination or repeated infections ...

Response 1: Thank you for pointing this out comment. We agree with this comment. We replaced the word "during" with "after" (line 61, p.2)

Comments 2: Line 67 - "Characterized" may be a better word than "known".

Response 2: Thank you for pointing this out comment. We agree with this comment. We replaced the word "known" with "characterized" (line 68. p.2).

Comments 3: Line 97 - suggest: affinities are essentially equivalent

Response 3: Thank you for pointing this out comment. We agree with this comment. We have changed the wording. We replaced the word " affinity " with "affinities" (line 97. p.3)

Comments 4: General comment:  there is a great deal of information and data characterizing the activity of these antibody sets with constructed and expressed antigens. It would be beneficial to relate how this compares to and is relevant to antibody binding to native viral protein. Are there biological consequences (neutralization, etc.)?

Response 4: Thank you for pointing this out comment. We have added information to the text of the article.

It can be expected that the result obtained with recombinant proteins reflects the interaction of the studied antibody with the surface protein E in the virus. And it will bind with high affinity to viral particles in the body. Since 10H10 is known not to be neutralizing, the most understandable use for it may be the selective delivery of antiviral drugs. Affinity in the range of 10-9 M can ensure selective accumulation of the antibody conjugate with an antiviral substance at the site of virus concentration. (line 256-262 p.11)

Comments 5: Line 477 - suggest: delete "our experiments have shown" It is wordy and unnecessary.

Response 5: Thank you for pointing this out comment. We agree with this comment. We removed it. (line 496 p.17)

Comments 6: This approach has defined the binding structures very well. Are there biological consequences or implication from this that relate to infectivity or tissue tropism? Why was this study important?

Response 6: Thank you for pointing this out comment. We have added information to the text of the article.

Such a detailed study of the epitope structure is of fundamental importance because it allows us to provide a molecular basis for the cross-reactive nature of this antibody. A precise understanding of the amino acid residues involved in the formation of the complex allows us to state that this antibody can interact not only with the proteins of the flaviviruses used in the work, but also with a large number of others that have similar amino acid sequences of the fusion loop. In an applied sense, this work is important as a basis for changing the structure of antibodies to obtain a chimeric or humanized variant that will have reduced allergenicity. The modified antibody can become a platform for creating a pan-flavivirus antiviral drug. (line 509-518 p.17-18)

Reviewer 3 Report

Comments and Suggestions for Authors

The authors investigated the interaction between the mouse monoclonal antibody (mAb) 10H10 and viruses of the genus Orthoflavivirus, which are pathogenic to humans. Their study focused on mapping the epitope of this antibody, which is the specific part of the virus that the antibody recognizes. They used ELISA, molecular modeling, and phage display technology to identify that mAb 10H10 recognizes a conserved region (amino acids 98-110) in the E protein of flaviviruses like tick-borne encephalitis, Zika, and dengue viruses. The structural interactions between this region and the antibody were further analyzed using molecular docking and dynamics.

After reviewing the article, I find it clear and informative. The introduction effectively outlines the study's objectives and emphasizes the research's significance. The materials and methods section provides a detailed and accurate description of the experimental procedures. The results are presented clearly and are easy to follow.

However, I noticed that some figures were included in the discussion section rather than in the results section, which is unusual. I strongly advise the authors to move these figures to the results section and to revise the discussion accordingly to maintain consistency and clarity throughout the manuscript (Figure 6, Figure 7, and Figure 8).

Apart from this, the discussion interprets and integrates the findings effectively. Overall, this paper meets the criteria for publication. I recommend acceptance after the authors address the suggested revisions mentioned above.

Further detailed and specific comments are provided below.

Main Question Addressed by the Research

What is the specific epitope of the mouse monoclonal antibody 10H10 that interacts with human-pathogenic viruses of the genus Orthoflavivirus, and how can this be mapped using phage display technology?

Originality and Relevance to the Field

The research is original and relevant as it identifies a highly conserved epitope region in flaviviruses that mAb 10H10 interacts with. This addresses a specific gap in understanding the molecular interactions between antibodies and viral proteins, which is crucial for therapeutic and diagnostic developments.

Contribution to the Subject Area

This study adds to the existing knowledge by precisely mapping the epitope of mAb 10H10, identifying the conserved region (98-DRGWGNXXGLFGK-110) on the flavivirus E protein. This information is valuable for designing targeted therapies and vaccines against flaviviruses like TBEV, ZIKV, and DENV.

Methodology Improvements and Further Controls

The current methodology appears robust, using phage display, ELISA, molecular docking, and dynamics methods. No specific improvements or further controls are suggested.

Consistency of Conclusions with Evidence

The conclusions are consistent with the evidence presented. The research effectively addresses the main question posed by demonstrating that the identified conserved epitope region interacts with the mAb 10H10.

Appropriateness of References

Yes, the references are appropriate.

Additional Comments on Tables and Figures

The tables and figures are satisfactory and clearly present the data. I noticed that some figures were included in the discussion section rather than in the results section, which is unusual. I strongly advise the authors to move these figures to the results section and to revise the discussion accordingly to maintain consistency and clarity throughout the manuscript (Figure 6, Figure 7, and Figure 8). No additional comments.

Author Response

Thank you very much for taking the time to review this manuscript. Please find the detailed responses below and the corresponding revisions in the re-submitted files.

Comments 1: The tables and figures are satisfactory and clearly present the data. I noticed that some figures were included in the discussion section rather than in the results section, which is unusual. I strongly advise the authors to move these figures to the results section and to revise the discussion accordingly to maintain consistency and clarity throughout the manuscript (Figure 6, Figure 7, and Figure 8).

Response 1: Thank you very much for such a high evaluation of our article. We tried to transfer figures 6, 7 and 8 to the results chapter. We managed to transfer figure 6 (line 195-205.p. 8-9), but when transferring figures 7 and 8, both the logic and clarity of the narrative are violated, since these figures, and especially 8, are generalizing figures that visualize the comparison of the results obtained.